# BOLD signal response in primary visual cortex to flickering checkerboard increases with stimulus temporal frequency in older adults

Yuji Uchiyama[1]*, Hiroyuki Sakai[1], Takafumi Ando[1¤a], Atsumichi Tachibana[1¤b], Norihiro Sadato[2]

**1** Human Science Research Domain, Strategic Research Division, Toyota Central R&D Labs., Inc., Nagakute, Aichi, Japan, **2** Division of Cerebral Integration, Department of System Neuroscience, National Institute for Physiological Sciences, Okazaki, Aichi, Japan

¤a Current address: Human-Centered Mobility Research Center, National Institute of Advanced Industrial Science and Technology, Higashi Tsukuba, Ibaraki, Japan
¤b Current address: Department of Histology and Neurobiology, Dokkyo Medical University, Mibu, Shimotsugagun, Tochigi, Japan
* uchiyama@mosk.tytlabs.co.jp

**Data Availability Statement:** All relevant data are within the paper and its Supporting information files.

## Abstract

Many older adults have difficulty seeing brief visual stimuli which younger adults can easily recognize. The primary visual cortex (V1) may induce this difficulty. However, in neuroimaging studies, the V1 response change to the increase of temporal frequency of visual stimulus in older adults was unclear. Here we investigated the association between the temporal frequency of flickering stimuli and the BOLD activity within V1 in older adults, using surface-based fMRI analysis. The fMRI data from 29 healthy older participants stimulated by contrast-reversing checkerboard at temporal flicker frequencies of 2, 4, and 8 Hz were obtained. The participants also performed a useful field of view (UFOV) test. The slope coefficient of BOLD activity regarding the temporal frequency of the visual stimulus averaged within V1 regions of interest was positive and significantly different from zero. Group analysis in the V1 showed significant clusters with positive slope and no significant clusters with a negative slope. The correlation coefficient between the slope coefficient and UFOV performance was not significant. The results indicated that V1 BOLD response to a flickering visual stimulus increases as the stimulus temporal frequency increases from 2 to 8 Hz in older adults.

## Introduction

Many older adults have difficulty seeing brief visual stimuli which younger adults can easily recognize [1–3]. The difficulty among older adults interferes with activities of daily living [4], including walking without falls [5] and safe driving [6–8]. Training in seeing brief stimuli improved the difficulty [9, 10]. The improvement suggests that causes of the reduced temporal-processing ability are not only attributed to optical factors, such as decreased ocular transmittance [11], increased intraocular scattering [12], and increased ocular aberration [13] in the eyes, but also to the visual processing deterioration in neuronal pathways.

**Funding:** Toyota group companies provided funds for research to Toyota Central R&D Labs., Inc. The funders provided support in salaries for authors YU, HS, TA, and AT but did not have any additional role in the study design, data collection, and analysis, decision to publish, or preparation of the manuscript. The specific roles of these authors are articulated in the "author contributions" section.

**Competing interests:** I have read the journal's policy, and the authors of this manuscript have the following competing interests: YU, HS, TA, and AT were employed by Toyota Central R&D Labs., Inc. The authors have not applied for any patent related to this study. This does not alter our adherence to PLOS ONE policies on sharing data and materials except for those including the privacy and personal information of research participants.

One of the potential brain regions causing difficulty in the pathways may be the primary visual cortex (V1). Previous studies showed that V1 response to visual stimuli changed with age. V1 neuronal response in older monkeys was different from that of younger monkeys [14–16]. Functional neuroimaging studies in humans showed that V1 neural responses to visual stimuli in older adults were lower than those in younger adults [17–20]. These results suggest that V1 is a potential region that causes the older adults' vulnerability to brief visual stimuli.

The difficulty of seeing brief visual stimuli in older adults is considered to be due to low neural response to brief visual stimuli. A low neural response can be considered low information processing ability in the brain. If some information processing ability in V1 becomes lower to the brief visual stimuli in older adults, V1 response (i.e., functional neuroimaging signal) to brief visual stimuli should be low. This means that the V1 response to flickering visual stimuli with higher temporal frequency is lower in older adults.

Functional neuroimaging studies of younger adults repeatedly showed that an increase of visual stimulus frequency of up to 8 Hz increases neuroimaging signal response in the V1 [21–29]. However, in older adults, the relation between the temporal frequency of the visual stimulus and the V1 response has not been clear. A positron emission tomography (PET) study by Mentis et al. demonstrated that cerebral blood flow (CBF) response to flickering stimuli increased as stimulus frequency increases in older adults [30]. An fMRI study by Cliff et al. suggested that the V1 BOLD response decreased with the increase of frequency in older adults. This result seems to be our expectation of V1 response in older adults. Another fMRI study by Fabiani et al. indicated a tendency of the increase in V1 BOLD response in older adults, although the temporal-frequency effect on BOLD response was not statistically tested [31].

A possible reason for the inconsistent results in older adults might be V1 localization methods in the studies. The past two fMRI studies did not clearly indicate that the regions in which BOLD response increased with the stimulus frequency were placed in V1. The study by Cliff et al. [32] only showed axial activation maps of group analysis to localize V1 and did not describe brain coordinates of peak voxels. The activation map indicated that the decrease was in the posterior part of V1, which means that the region with the decrease was in the peripheral visual field. Moreover, another fMRI study by Fabiani et al. [31] investigated BOLD signal averaged within V1 determined by inspection of individual anatomical images and did not show the standard brain coordinates of the region. On the other hand, the PET study by Mentis et al. [30] clearly showed a Talairach location within V1 where CBF was increased with temporal frequency. Thus, the past fMRI studies are insufficient to conclude whether increase or decrease in V1 BOLD activity as the stimulus temporal frequency increases.

Cortical surface-based analysis is superior to voxel-based analysis to investigate the V1 BOLD activity [33–37]. The V1 map, based on the cytoarchitecture of postmortem human brains, was implemented in a software suite for surface-based analysis as a standard brain map [35]. Voxel-based analysis can also use a cytoarchitectonic map of V1 [38]; however, the surface-based analysis can determine the V1 more accurately than the voxel-base analysis [37]. The reason for accuracy difference between the two analyses has been considered in that the cerebral cortical folding patterns (i.e., gyri and sulci) are effective to estimate V1 location. Additionally, surface-based analysis is more accurate for localizing the activated region and has higher efficiency for detecting activated region than voxel-based analysis [33, 34, 36]. Cortical surface is complexly folded with sulci and gyri; however, voxel-based analysis cannot possibly separate the neural activities between adjacent gyri like regions around the calcarine sulcus in V1. Thus, surface-based analysis is suitable for V1 activity studies.

This study aimed to investigate the association between the frequency of flickering stimuli and the BOLD activity within V1 in older adults, using surface-based fMRI analysis. The surface-based analysis will reveal the V1 region related to a temporal frequency change of visual

stimuli more accurately than the voxel-based analysis used in previous studies [30–32]. fMRI data from 29 healthy older participants were derived from contrast-reversing checkerboard stimuli at temporal flicker frequencies of 2, 4, and 8 Hz, and the resulting BOLD within V1 was examined. Furthermore, the factors associated with the individual difference of the V1 BOLD response were investigated. We examined the correlation between individual characteristics, including visual processing speed (i.e., useful field of view [UFOV] task [39] performance), and BOLD response change with the frequency.

## Materials and methods

### Participants

In this fMRI experiment, 15 male and 14 female participants (mean age, 68 years; age range, 65–74 years) were included. The following inclusion criteria were assessed using a questionnaire: absence of history of eye disease with normal and corrected-to-normal vision, absence of neurological or psychiatric illness, and absence of developmental disorders; a score of $\geq 24$ on the Mini-Mental State Examination [40]. All participants showed no pathological changes in their structural MRI images (reviewed by one experienced radiologist, NS) and were right-handed according to the Edinburgh Handedness Inventory [41].

The protocol was approved by the ethics committees of the National Institute of Physiological Sciences and Toyota Central R&D Labs., Inc. (Nagakute, Aichi, Japan). A written informed consent was obtained from all participants. This study was conducted according to the tenets of the Declaration of Helsinki.

### Imaging setup

Visual stimuli were delivered using Presentation software (Neurobehavioral Systems, Berkeley, CA, USA) on a computer CF-B10CWADR (Panasonic, Osaka, Japan). An LCD projector CP-SX12000 (Hitachi, Tokyo, Japan) projected the stimuli (resolution, 1280 × 1024 pixels) on a screen in an MRI scanner room. Participants were asked to lay down in a 3 T MRI MAGNETOM Verio (Siemens, Munich, Germany) with a 32-channel head coil, and visual stimuli were observed through a mirror mounted on a head coil. The viewing distance between the eye and the screen (size, 56.8 × 42.6 cm) was approximately 180 cm. A video camera captured the eyes of the participant through the mirror. The eye images were displayed in an MRI scanning control room. A USB data acquisition unit, USB-6009 (National Instruments, Austin, TX, USA), was used to synchronize the presentation of the visual stimuli with the fMRI scan.

### fMRI procedure

V1 neural activity was measured using a block-design fMRI procedure consisting of alternating rest block and stimulus block of 16 s each. In the stimulus block, a checkerboard pattern and its inverted pattern were alternately presented. The temporal flicker frequency of alternation was 2, 4, or 8 Hz, and the duty cycle of the alternation was 50%. The checkerboard pattern consisted of black and white squares (20 × 20) with a Michelson contrast of 0.97 (black: 6.77 cd/m$^2$; white: 500 cd/m$^2$) subtended approximately at 13.5 × 13.5˚. A red circle (subtending a visual angle of 0.27˚ at the diameter) was overlaid on the stimulus center as a fixation point. In the rest of the blocks, the same fixation point was presented on a uniform gray background. Three runs, each consisting of six stimulation blocks and seven rest blocks with a total of 208 s duration, were performed. One of the three flicker frequencies of 2, 4, and 8 Hz was randomly assigned to each of the three consecutive stimulus blocks. The frequency order in a run was randomized for each run, which means that the frequency orders were different among

participants. Participants were instructed to gaze at the fixation point throughout a run. During the checkerboard viewing task, experimenters kept watching the participants' eyes with a monitor displaying the eyes and confirming that the eyelids were open.

## MRI data acquisition

For fMRI runs, a T2*-weighted gradient-echo echo planar imaging (EPI) sequence was used. An EPI image consisted of 31 transaxial slices of 3-mm thickness each with no gaps. The slices were acquired in an ascending mode with oblique scanning. The following settings were used: repetition time (TR), 2000 ms; flip angle (FA), 76˚; echo time (TE), 30 ms; field of view (FOV), 192 mm; and in-plane matrix, 64 × 64 pixels. A total of 107 EPI images were acquired for each run. A structural brain image was acquired for each participants using T1-weighted magnetization-prepared rapid gradient-echo imaging sequences with the following settings: TR, 1800 ms; TE, 2.97 ms; FA, 9˚; FOV, 250 × 250 mm; in-plane matrix size, 256 × 256 pixels; slice thickness, 1 mm; 192, contiguous transaxial slices).

## Visual processing speed task

The study participants performed a UFOV test before the fMRI data acquisition. A participant performed 288 test trials. In the trial, a visual stimulus was presented with a short duration selected randomly from 40, 60 100, 180, 340, and 660 ms. At the center of the visual stimulus, a randomly selected letter from E, F, H, and L was placed. In each of the four peripheral corners around the letter, a circle was placed. One randomly selected circle from the four peripheral corners was filled, and the other three were opened. After the stimulus was presented, the participant responded as to which letter was presented and which circle was filled. The correct trials were that both responses to the center and the peripheral stimulus were correct. The UFOV test performance of the participant was stimulus duration of 53% correct rate of the responses estimated from all trial responses of the participant. To estimate the duration, a logistic-regression model was fitted to the UFOV trial response with respect to the stimulus duration. Using the model, the stimulus duration of 53% correct rate of the response was estimated; thus, shorter stimulus duration represents higher UFOV performance. The detailed task and analysis procedures have been described elsewhere [39].

## fMRI data analysis

We conducted two fMRI analyses: categorical analysis which shows BOLD activity against the resting baseline for each stimulus frequency (i.e., 2, 4, and 8 Hz) and parametric analysis which shows the BOLD activity change regarding the stimulus frequency increase (i.e., slope) and the averaged BOLD response to the stimulus block regardless of its stimulus frequency (i.e., offset). We performed both analyses using the FreeSurfer software suite (version 7.7.1) [42] on Linux (CentOS 7). The first step of preprocessing was common to the two analyses.

**Preprocessing.** Brain surfaces were segmented from T1 images of each participant. If the dura mater were included in the surface image, we edited the skull-stripped images (i.e., brainmask.nii.gz) manually to exclude the dura mater, and the surface images were corrected. The brain surfaces of the left and right hemispheres, derived from T1 images of each participant, were tessellated with vertices and were inflated without stretching. The inflated surfaces were projected on a standard sphere of the brain using information of the gyri and sulci. Based on the mapping information between the standard sphere and individual sphere, individual V1 region was decided using the cytoarchitectonic V1 map (i.e., vcAtlas) [35].

Brain activity was analyzed using FsFast (FreeSurfer Functional Analysis Stream) tools. The first three EPI images of each run were discarded to allow for stabilization of the magnetic

field. All the remaining EPI images were corrected for slice timing and head motion. The corrected EPI images were registered on the surfaces of individual and of standard averaged brain (i.e., fsaverage). EPI signal mapped on the surfaces were smoothed using a two-dimensional Gaussian kernel with a full width at half-maximum of 5 mm.

## Categorical fMRI analysis of visual stimulus frequency

**Individual analysis.** For individual level analysis, the expected BOLD activity for each run and for each flicker frequency was modeled as a regressor of interest using the boxcar function, convolved with a hemodynamic response function (HRF), which was SPM canonical without derivatives. The regressors were fitted to the EPI time series for each vertex on individual and standard averaged surface. Estimates obtained by the fit were considered to represent the BOLD activity for each flicker frequency.

**Regions of interest (ROI) analysis.** To analyze the overall V1 activity, we conducted most ROI analyses using Matlab 2019a with Statistics and Machine Learning Toolbox (Mathworks Inc.). Mauchly's test of sphericity and one-way repeated measures analysis of variance (ANOVA) were conducted using anovakun 4.8.5 software package [43] on R 4.0.5. Vertices within the ROIs satisfied the following two conditions: 1) significantly activated by the stimulus with surface-based analysis (uncorrected $P < 0.05$) and 2) within V1 defined by vcAtlas for each participant. The magnetic resonance (MR) signal changes were averaged across the ROI for each frequency and each participant (S1 File). After Mauchly's test of sphericity, repeated-measured one-way ANOVA, with frequency as the within-subjects factor, was applied to the averaged MR signal changes. Tukey's tests were performed as post-hoc tests of the ANOVA. Statistical significances of the sphericity test, the ANOVA and the post-hoc tests were set at $P < 0.05$. To confirm the increase of the MR signal change within the ROI with the frequency, Pearson's product-moment correlation coefficients between the frequency and MR signal change within the ROI were tested for their difference from zero by one-sample t-test. The significance level of the t-test was $P < 0.05$.

**Group analysis.** To compare the brain activity among all participants between the frequencies, group level analysis of the brain activity on an averaged brain surface was conducted using permutation tests using FsFast tools. The following three contrasts were tested: 1) brain activity at 8 Hz was higher than that at 2 Hz (8 Hz > 2 Hz), 2) brain activity at 4 Hz was higher than that at 2 Hz (4 Hz > 2 Hz), and 3) brain activity at 2 Hz was higher than that at 8 Hz (2 Hz > 8 Hz). Significance threshold of the test was set at cluster level corrected $P < 0.05$.

## Parametric fMRI analysis

**Individual analysis.** The slope and offset contrasts for each run were modeled as two regressors. The slope regressor was a boxcar function whose amplitude was modulated by the stimulus frequency. The offset regressor was a boxcar function whose stimulus onset and offset corresponded to those of stimulus blocks, respectively, with constant amplitude regardless of the frequency. Both regressors were convolved with a HRF, SPM canonical without derivatives. The regressors were fitted to the EPI time series for each vertex on individual and standard averaged surface.

**Regions of interest (ROI) analysis.** To analyze the overall slope within V1, we conducted ROI analyses using Matlab 2019a with Statistics and Machine Learning Toolbox (Mathworks Inc.). Vertices within the ROIs satisfied the following two conditions: 1) the offset contrast was significant and 2) within V1 defined by vcAtlas for each participant. We averaged the effect size of the slope across the ROI for each participant (S2 File). The averaged effect size of the

slope and the offset within the ROI were tested for the difference from zero across the participants using a one-sample t-test. The significance level was set at $P < 0.05$.

To investigate the individual difference of the V1 BOLD response to the visual stimulus frequency, we tested Pearson's product-moment correlation coefficient between the effect size of the slope or the offset within the ROI and the individual characteristics of age, MMSE, or UFOV performance. The significance level was set at $P < 0.05$ for the correlation test.

**Group analysis.** A group level analysis of the positive and negative contrasts of the slope and the offset on an averaged brain surface was conducted using permutation tests with FsFast tools. The significance threshold of the test was set at cluster level corrected $P < 0.05$.

## Results

### Categorical fMRI analysis of visual stimulus frequency

**Individual analysis.** This study first explored MR signal change within V1 for each participant. Fig 1 shows the highest peak MR signals in the left and right V1 among the three representative subjects. The peaks were found in and around the calcarine sulcus within V1. The MR signal on the peaks increased with the temporal frequency for each hemisphere and each participant. Among all 29 participants, 13 participants showed that peak MR signal increase with the stimulus frequency (i.e., the order of MR signal change is 8 Hz > 4 Hz > 2 Hz, which is identical of Fig 1) in V1 of both hemispheres, and three participants showed that the peak MR signal increase in V1 of one hemisphere.

**ROI analysis.** The association between MR signal on V1 and the temporal frequency in the ROI was examined. The MR signal averaged in V1 increased with the temporal flicker frequency (Fig 2 and Table 1). A repeated-measured one-way ANOVA was performed to test the effect of the frequency on the MR signal in the ROI. Mauchly's test of sphericity was accepted ($\chi^2 (2) = 4.53$; $P = 0.10$). The ANOVA revealed a significant effect of the temporal frequency on the MR signal averaged across the ROI ($F(2,56) = 15.1$; $P = 5.52 \times 10^{-6}$, $\eta_P^2 = 0.35$). A post-hoc analysis using Tukey's test indicated that the MR signal changes of 4 Hz was larger than that of 2 Hz ($P = 0.0012$, 95% CI = 0.05–0.22) and that of 8 Hz was larger than that of 2 Hz ($P = 0.00017$, 95% CI = 0.11–0.34). However, the MR signal changes of 8 Hz was not significantly larger than that of 4 Hz ($P = 0.082$, 95% CI = −0.01–0.19). Additionally, to confirm the increase of the MR signal in the ROI with the temporal frequency, one-sample t-test of the mean correlation coefficients between the frequencies and mean MR signal averaged across the ROI was performed. The mean correlation coefficient (mean R = 0.63, SD = 0.56) was significantly larger than zero (one-sample t-test, t(28) = 6.07, $P = 1.50 \times 10^{-6}$, 95% CI = 0.42–0.85). These results showed that the V1 BOLD activity increased with the temporal flicker frequency.

**Group analysis.** Group analysis was conducted to validate that V1 activity increases with the temporal frequency. The stimulus with each temporal frequency significantly induced V1 brain activity on both hemispheres (cluster-wise corrected $P < 0.05$) (Fig 3A and 3B, and Table 2). Furthermore, similar relations obtained by the post-hoc tests of the ROI analysis were confirmed. V1 activity at 8 Hz was significantly higher than that at 2 Hz (cluster-wise corrected $P < 0.05$) (Fig 3C and Table 2). V1 activity at 4 Hz was also significantly higher than that at 2 Hz (cluster-wise corrected $P < 0.05$) (Fig 3D and Table 2).

To find that V1 region decreased with the temporal frequency, the contrast between 2 Hz > 8 Hz was tested. This study could not find any V1 region in which activity at 2 Hz was larger than that at 8 Hz.

Finally, this study checked that the peak positions of the individual analysis of the three representative subjects shown in Fig 1 were on the group analysis results of the 8 Hz > 2 Hz

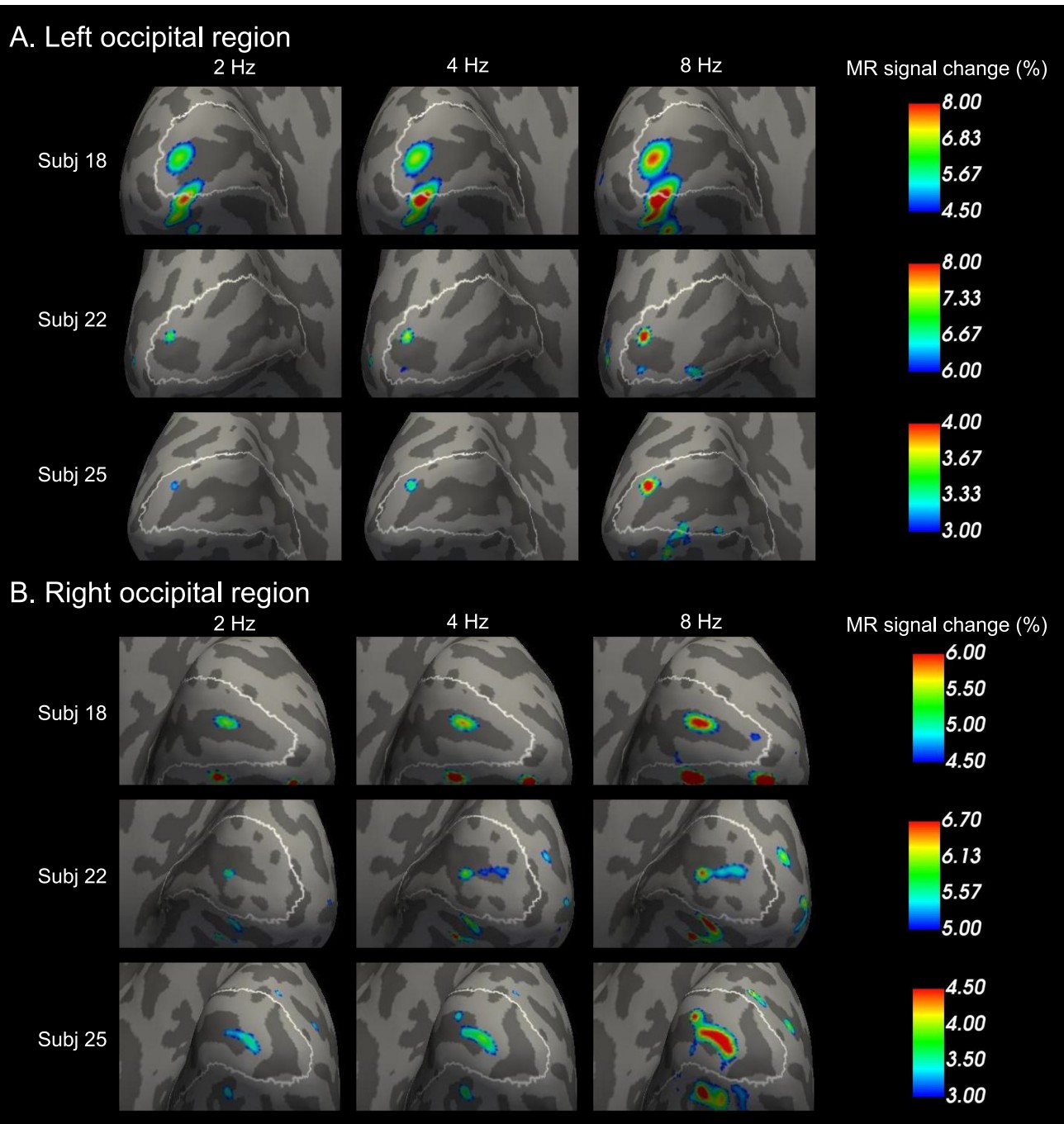

**Fig 1. Peak BOLD activities in V1 increase as temporal flicker frequency increases (2, 4, and 8 Hz) in three representative participants.** (A) MR signal changes are overlaid on an inflated brain surface of each participant in the left occipital region. The brain surfaces are shown from a posterior/medial viewpoint. White lines on the brain surfaces indicate the border of V1, based on cytoarchitecture using vcAtlas [35]. Dark and light gray regions represent sulci and gyri, respectively. Calcarine sulci are dark gray regions passing near the center of V1 and extending from anterior to posterior within the region bordered by the white lines (i.e., V1). Peaks of MR signal change within the region bounded by the white lines (i.e., V1) increase from the left (i.e., 2 Hz) to the right (i.e., 8 Hz). (B) Results in the right occipital region correspond to (A).

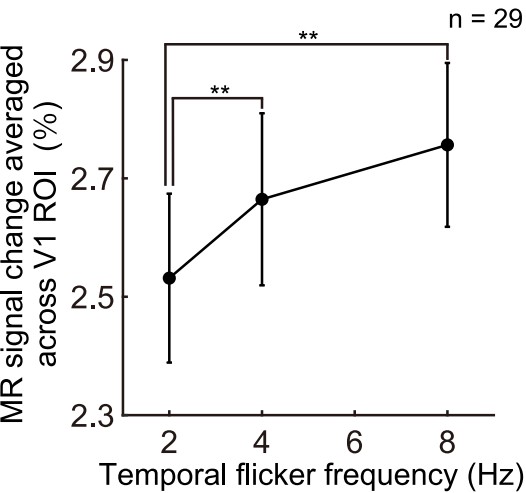

**Fig 2. Magnetic resonance (MR) signal change in V1 increases with temporal flicker frequency.** The MR signal change was averaged across V1 regions of interest (ROIs) for each participant and across 29 participants. The V1 ROIs were cortical vertices activated significantly (vertex-level uncorrected $P < 0.001$) within the V1 based on cytoarchitectonic atlas. Error bars represent the standard errors of the MR signal change. The main effect of the flicker frequency in one-way analysis of variance was found to be significant ($P = 6.5 \times 10^{-6}$). **: $P < 0.01$ (Tukey's test).

contrast in Fig 3C. Fig 4 shows that every peak shown in Fig 1 were in regions overlapping between V1 and the region of 8 Hz > 2 Hz.

## Parametric fMRI analysis

**Individual analysis.** This study has examined the slope and the offset of the BOLD activity within V1 for each participant. Fig 5 shows the highest peak of the slope and the offset in the left and right V1 among the three representative subjects. The peaks were found in and around the calcarine sulcus within V1.

**ROI analysis.** This study confirmed the increase of the BOLD activity in the ROI with the temporal frequency using the slope coefficient. In one participant, any vertex, which met the ROI inclusion criteria, was not found. Thus, we performed the analysis with 28 participants. The BOLD slope and offset coefficients averaged within the ROI were tested across the 28 participants using a one-sample t-test. The mean slope coefficient (mean coefficient = 0.049, SD = 0.56), and the mean offset coefficient (mean coefficient = 2.45, SD = 0.78) were significantly larger than zero (one-sample t-test, t(27) = 5.94, $P = 2.38 \times 10^{-6}$, 95% CI = 0.032–0.065, and t(27) = 16.5, $P = 1.27 \times 10^{-15}$, 95% CI = 2.14–2.75, respectively). The positive slope coefficient showed that the V1 BOLD activity increased with the temporal flicker frequency.

**Table 1. Magnetic resonance (MR) signal change averaged within each region of interest and across all participants (n = 29).**

| Temporal flicker frequency (Hz) | MR signal change (%) Mean (SD) |
|:---:|:---:|
| 2 | 2.53 (0.77) |
| 4 | 2.66 (0.78) |
| 8 | 2.75 (0.75) |

SD: standard deviation.

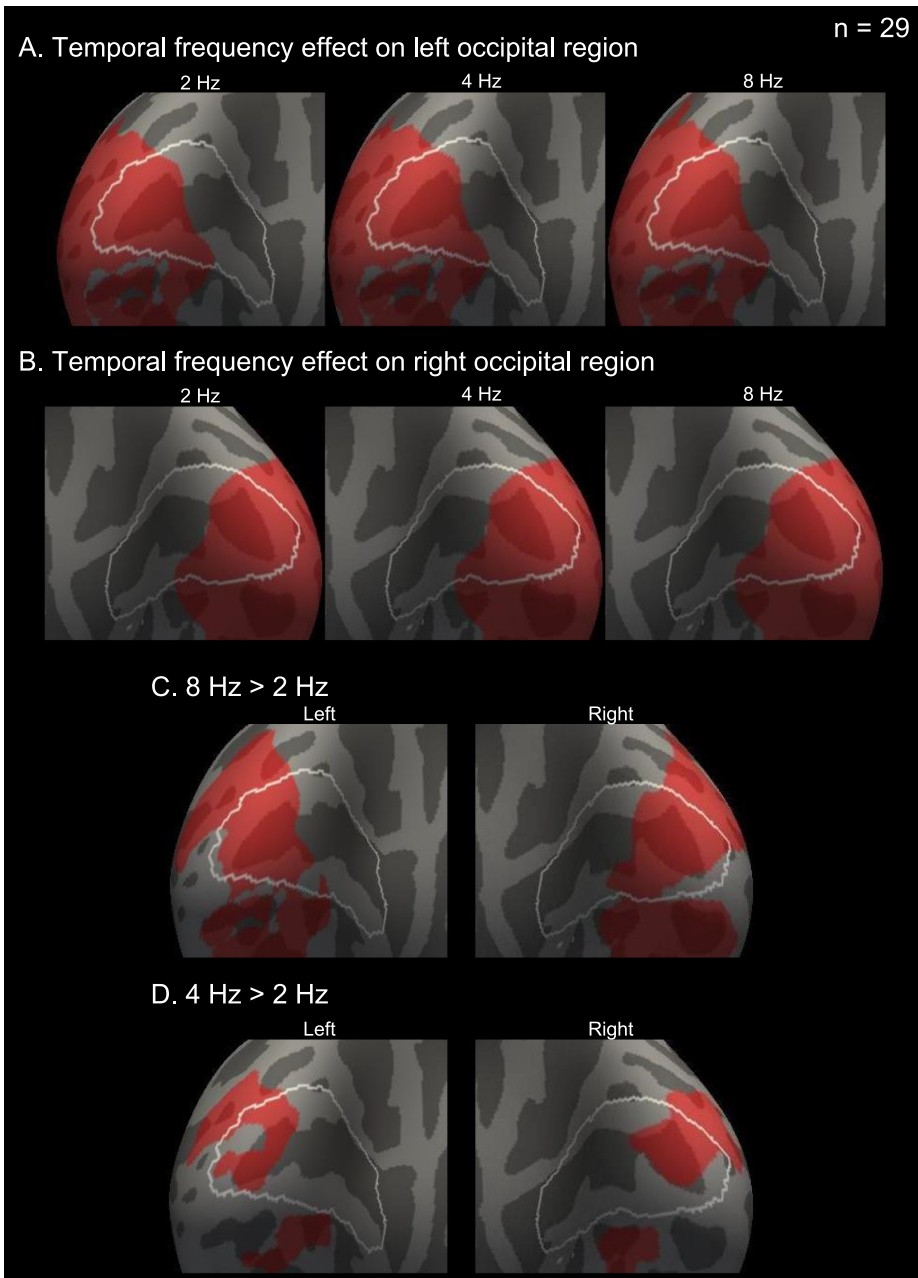

**Fig 3. Significant brain activity derived from group analysis (n = 29).** Red regions indicate statistically significant clusters activated by each temporal flicker frequency (2, 4, and 8 Hz) on the left (A) and right (B) hemispheres. (C) Statistically significant clusters in which the brain activity of 8 Hz is higher than that of 2 Hz (8 Hz > 2 Hz contrast). (D) Statistically significant clusters in which the brain activity of 4 Hz is higher than that of 2 Hz (4 Hz > 2 Hz contrast). Brain surfaces are shown from a posterior/medial viewpoint. White lines on the brain surfaces indicate the border of V1. Statistical significance is cluster-wise corrected $P < 0.05$.

**Group analysis.** This study confirmed the brain areas in which the BOLD activity increased with the slope contrast of the group analysis. In the positive slope contrast, significant regions were found in the posterior part in the V1 on both hemispheres (cluster-wise corrected $P < 0.05$) (Fig 6A, and Table 3). In the negative slope contrast, there were no significant clusters in the V1 on both hemispheres.

**Table 2. Statistically significant clusters for each contrast by the group analysis (n = 29).**

| Contrast | Side | Cluster | | | Maximum peak vertex | | | |
|---|---|---|---|---|---|---|---|---|
| | | Cluster-wise corrected *P* | (mm²) | Cortical region | Vertex-level uncorrected *P* | MNI coordinates | | |
| | | | | | | x | y | z |
| 2 Hz | Left | 0.003 | 6132.02 | Lateral occipital | $3.3 \times 10^{-18}$ | -12.3 | -102.0 | 3.6 |
| | Right | 0.003 | 7160.79 | Lateral occipital | $3.5 \times 10^{-16}$ | 14.9 | -101.3 | 5.9 |
| 4 Hz | Left | 0.003 | 5097.11 | Lateral occipital | $6.9 \times 10^{-18}$ | -13.7 | -101.7 | 3.9 |
| | Right | 0.003 | 6037.11 | Lateral occipital | $9.8 \times 10^{-16}$ | 17.0 | -100.9 | 3.7 |
| 8 Hz | Left | 0.003 | 6626.71 | Lingual | $3.0 \times 10^{-18}$ | -9.0 | -92.6 | -10.0 |
| | Right | 0.003 | 7322.04 | Lateral occipital | $4.9 \times 10^{-16}$ | 15.4 | -101.0 | 6.0 |
| 8 Hz > 2 Hz | Left | 0.003 | 3724.19 | Lateral occipital | $4.2 \times 10^{-10}$ | -20.6 | -97.4 | 6.5 |
| | Right | 0.003 | 3707.5 | Lateral occipital | $6.7 \times 10^{-10}$ | 14.1 | -99.0 | 11.6 |
| 4 Hz > 2 Hz | Left | 0.003 | 885.56 | Lateral occipital | $1.1 \times 10^{-6}$ | -14.6 | -101.0 | 4.7 |
| | Right | 0.003 | 968.41 | Lateral occipital | $1.4 \times 10^{-7}$ | 19.1 | -97.3 | 14.9 |

MNI: Montreal Neurological Institute.

Note: Cortical regions are indicated by Desikan-Killiany atlas [44].

Additionally, the study examined the positive and negative offset contrast. The significant positive and negative areas were on the posterior and the anterior part in the V1 on both hemispheres, respectively (Fig 6B and Table 3).

**Effect of individual characteristics on BOLD response.** To investigate the individual difference of V1 BOLD response to the stimulus frequency, we examined the correlation between

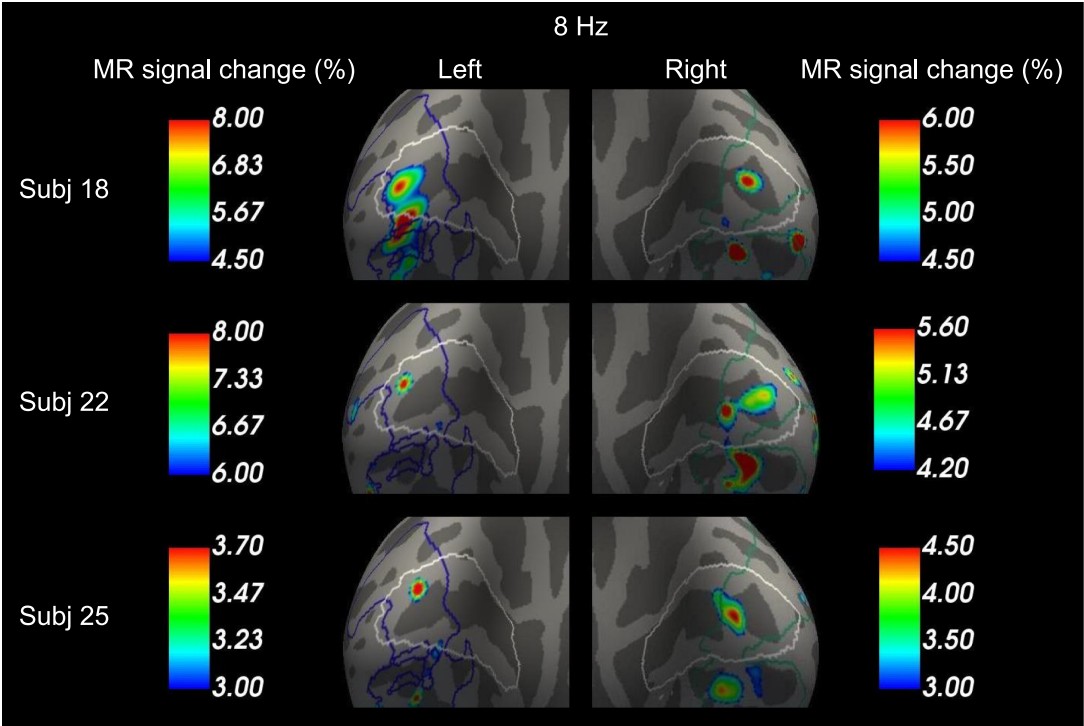

**Fig 4. Individual peak responses to 8 Hz are within the V1 region in which the brain activity of 8 Hz was higher than that of 2 Hz in the group analysis.** Peak brain responses to 8 Hz among the three representative subjects in Fig 1 are overlaid on the group analysis results of 8 Hz > 2 Hz contrast in Fig 3C. Blue and green lines on the brain surfaces indicate the border of the 8 Hz > 2 Hz region. White lines on the brain surfaces indicate the border of V1.

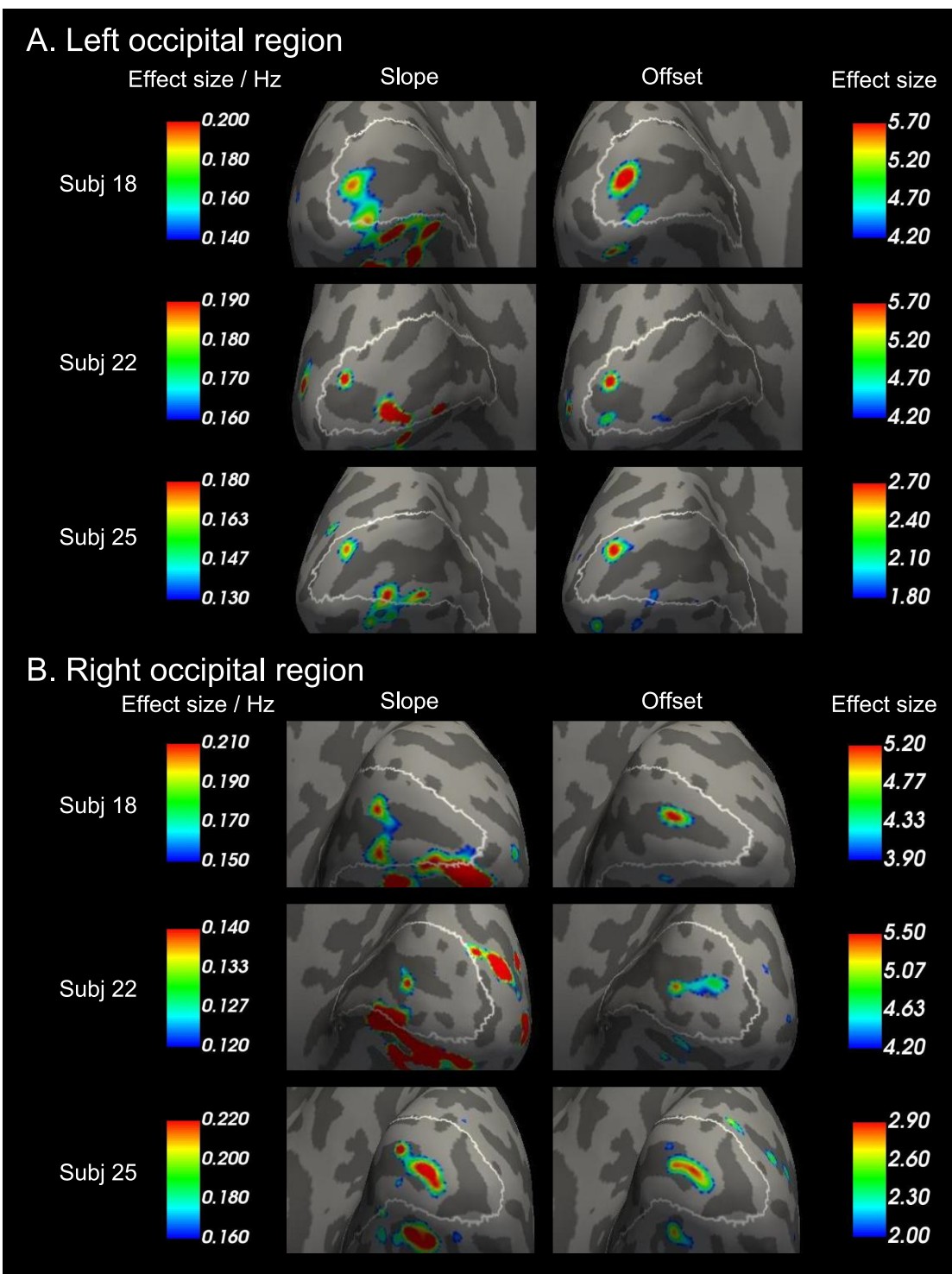

**Fig 5. Peak slope and offset in V1 among the three representative participants.** (A) Peak effect sizes of slope and offset contrasts are overlaid on an inflated brain surface of each participant in the left occipital region. The brain surfaces are shown from a posterior/medial viewpoint. White lines on the brain surfaces indicate the border of V1, based on cytoarchitecture using vcAtlas [35]. Dark and light gray regions represent sulci and gyri, respectively. Peaks of the slope and the offset within the region bounded by the white lines (i.e., V1). (B) Results in the right occipital region correspond to (A).

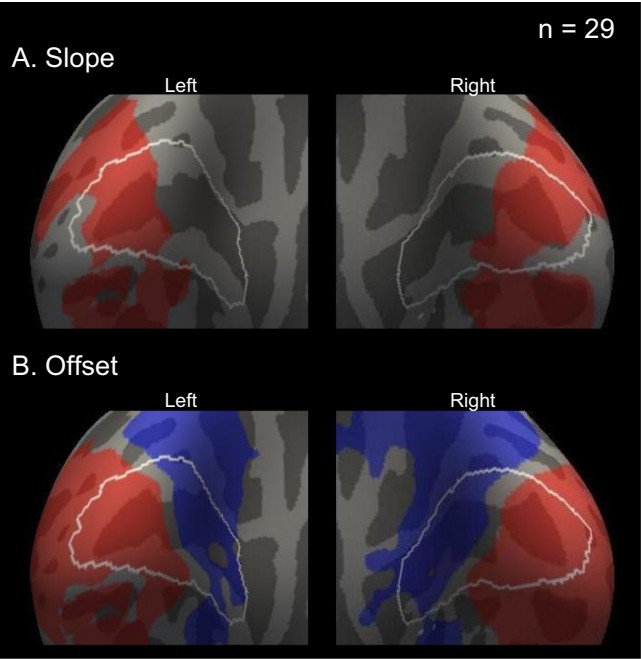

**Fig 6. Significant regions of slope and offset by group analysis (n = 29).** The significant slope (A) and offset clusters (B) were overlaid on the standard inflated brain surface. Red and blue regions indicate positive and negative contrast, respectively. Brain surfaces are shown from a posterior/medial viewpoint. White lines on the brain surfaces indicate the border of V1. Statistical significance level was set at cluster-wise corrected $P < 0.05$.

the BOLD slope or offset coefficients obtained by the parametric ROI analysis and the individual characteristics of age, MMSE, or UFOV test performance (Table 4). All of the correlations were not significant.

## Discussion

This study has investigated how the temporal frequency of flickering checkerboard stimuli modulate V1 BOLD activity in older adults, using surface-based analysis. The categorical analysis showed that the BOLD activity increases with flicker frequency from 2 to 8 Hz within the V1 region activated by visual stimuli. The one-way ANOVA of BOLD activity in the V1 ROI

**Table 3. Summary of statistically significant clusters for the slope and offset contrasts by group analysis (n = 29).**

| Contrast | Side | Cluster | | | Maximum peak vertex | | | |
|---|---|---|---|---|---|---|---|---|
| | | | | | Vertex-level uncorrected $P$ | MNI coordinates | | |
| | | Cluster-wise corrected $P$ | (mm$^2$) | Cortical region | | x | y | z |
| Positive slope | Left | 0.003 | 3840.12 | Lateral occipital | $1.0 \times 10^{-9}$ | -20.9 | -97.0 | 6.5 |
| | Right | 0.003 | 3922.04 | Pericalcarine | $3.5 \times 10^{-9}$ | 12.1 | -93.6 | 8.4 |
| Positive offset | Left | 0.003 | 4938.82 | Lateral occipital | $1.7 \times 10^{-16}$ | -15.3 | -100.8 | 4.5 |
| | Right | 0.003 | 6488.09 | Lateral occipital | $1.2 \times 10^{-15}$ | 16.1 | -100.5 | 7.0 |
| Negative offset | Left | 0.003 | 3281.61 | Cuneus | $2.9 \times 10^{-10}$ | -5.6 | -86.2 | 19.2 |
| | Right | 0.003 | 5168.37 | Cuneus | $2.9 \times 10^{-10}$ | 7.1 | -81.5 | 16.7 |

MNI: Montreal Neurological Institute.

Note: Cortical regions are indicated by Desikan–Killiany atlas [44]. Significant clusters with negative slope were not found.

**Table 4. Correlation coefficients between BOLD responses by parametric analysis and individual characteristics (n = 28).**

| Variables | Mean (SD) | Slope | | Offset | |
|---|---|---|---|---|---|
| | | R (95% CI) | P | R (95% CI) | P |
| Age | 68.4 (3.0) | 0.09 (-0.29–0.45) | 0.63 | 0.06 (-0.42–0.32) | 0.77 |
| MMSE | 28.4 (1.7) | 0.03 (-0.34–0.40) | 0.87 | 0.34 (-0.04–0.63) | 0.08 |
| UFOV | 336.0 (260.0) | -0.16 (-0.50–0.23) | 0.41 | 0.01 (-0.37–0.38) | 0.97 |

SD: standard deviation, R: Pearson's product-moment correlation coefficients, CI: confidence interval, MMSE: Mini-Mental State Examination, UFOV: useful field of view.

Note: MMSE represents the total score of MMSE test, while UFOV represents UFOV test performance which is visual stimulus duration in ms when the correct response rate is 53%.

showed a statistically significant main effect in flicker frequency. Post-hoc ANOVA test showed that BOLD activity of 8 Hz in the ROI was higher than that of 2 Hz and that BOLD activity of 4 Hz was higher than that of 2 Hz. The correlation coefficients between the temporal frequency and V1 BOLD activity were found to be significantly positive. Group analysis showed similar results of the post-hoc test; significant BOLD activity of 8 Hz > 2 Hz and 4 Hz > 2 Hz contrasts were found in V1. Conversely, any regions in which BOLD activity of 2 Hz was significantly higher than that of 8 Hz could not be found in V1. The parametric analysis showed the same relation derived by the categorical analysis. Slope coefficient averaged in the V1 ROI was positive and statistically different from zero. Group analysis showed significant clusters with a positive slope and no significant clusters with negative slope, in the V1.

This study is considered to locate BOLD response and V1 more precisely than the past studies using voxel-based analysis [30–32]. Surface-based analysis possibly located the region in which BOLD activity increased with the temporal frequency more precisely than those of the past functional imaging studies which used voxel-based analysis [30–32]. The surface-based analysis can separate the brain activity around folded brain surfaces [33, 34, 36]. Furthermore, V1 in this study is located more precisely than in the past fMRI studies [31, 32]. This study used a template atlas based on cytoarchitecture of postmortem brains (i.e., vcAtlas) to localize V1. The atlas well matched the V1 which was retinotopically defined [35]. Inversely, the past fMRI studies [31, 32] defined V1 based on human observation.

Our results are consistent with the PET study by Mentis et al. [30]. The study demonstrated that V1 activity significantly increased with temporal stimulus frequency in older adults. The location in which Mentis et al. found the increase in V1 activity was close to the peak position of MR signal in our results of the individual and the group analysis. The locations of both PET study and this study were around the posterior part of the calcarine sulcus. These regions were near the center of the visual field, in which flickering visual stimuli were presented.

Our results are inconsistent with the fMRI study by Cliff et al. [32], who suggested that the V1 brain region in which the response of BOLD fMRI activity to flickering checkerboard stimuli decreased with its frequency in older adults [32]. We could not find any significant V1 region in which the BOLD signal decreased with flicker frequency from the comparison between the BOLD activity at 2 and 8 Hz.

The results of this study suggest that the difference between the V1 activity changes with the temporal frequency of younger and older adults may be low. This study showed that the V1 activity increased with the temporal frequency from 2 to 8 Hz in older adults. Previous studies among younger adults [21–29] also showed the increase in V1 brain response with stimulus temporal frequency up to 8 Hz. According to the present and the previous results, it

is expected that the temporal-frequency dependence of V1 activity up to 8 Hz may be similar between younger and older adults.

Negative BOLD activities in V1 were found in the offset contrast by the parametric group analysis (Fig 6B). The regions with negative BOLD activities were adjacent to the regions with positive BOLD activities within V1. Past human fMRI studies repeatedly showed the same pattern of V1 BOLD activity [45–47]. In addition, monkey electrophysiological studies confirmed this spatial distribution of neuronal activity in V1 [48]. The negative BOLD activity of the offset contrast in this study is consistent with the past studies [45–48]. These facts support that our experiment and analysis were appropriate.

This study has two limitations. In order to match the study design to the previous studies [30–32], this study used a passive viewing task which did not require participants to perform a vigilance task during flicker stimulus viewing. The passive viewing task might induce sleepiness in participants and might affect V1 BOLD response. A recent study about BOLD response change with temporal frequency of flickering visual stimulus in younger adults used vigilance task during stimulus viewing and excluded participants who did not meet the criteria of vigilance task performance [21]. Although the task in this study was passive, the experimenter monitored that the eyelids of the participants were open during the task. Thus, this limitation may be a potential concern in our study. Future studies are required to use a vigilance task and to include participants who meet the criteria of vigilance task performance. The other limitation is that fMRI analysis used the canonical HRF without derivatives. This assumes that the HRF does not change with age. However, the HRF is changed with age [49] and unfitted HRF decreases the statistical power and the effect size of BOLD signal change [50]. Use of appropriate HRF according to older adults may derive higher BOLD activity than this study.

The correlation coefficient between the BOLD slope and the UFOV test performance was negative, although the correlation coefficient was not significant (Table 4). This negative correlation means that the participants with the lower UFOV test performance have a lower BOLD slope coefficient in V1. This negative correlation tendency corresponds to the expectation that the participants who are vulnerable to brief visual stimuli have low V1 response to the higher frequency. The improvement of the vigilance control during the task and using an appropriate HRF for data analysis, as discussed previously, might help to get a clearer relation.

Contrary to our expectation that V1 BOLD activity in older adults decreases when the visual stimulus frequency increases, all results of the group analysis showed no V1 region wherein the BOLD response decreased with frequency. The results of the categorical analysis showed that no V1 region wherein the 2 Hz BOLD activity were higher than the 8 Hz BOLD activity and the correlation coefficients between the frequency and the BOLD signal at the V1 ROI were significantly positive. Additionally, the parametric analysis results showed no V1 regions with negative slope coefficients. However, the correlation coefficient between the slope and UFOV test performance is negative, although no statistically significant difference was observed. This means that the participants with lower slope tended to have lower UFOV performance. This was consistent with our expectation. The above results present that the slope is positive for the older adults as a whole, however, individual differences in the slope among the older adults may be large and lower slope may impair the visual processing speed.

## Conclusion

In older adults, the V1 BOLD response to flickering visual stimulus increases as the temporal flicker frequency increases from 2 to 8 Hz.

## Supporting information

**S1 File. MR signal change in V1 ROIs.** MR signal change averaged across the V1 ROI for each participant (i.e., mri_segstats command output of hOc1). Table 1 was derived from this data.
(XLSX)

**S2 File. Effect size of the slope, and the offset in V1 ROIs, and data of individual characteristics.** The slope and offset effect size across the V1 ROI for each participant (i.e., mri_segstats command output of hOc1) and individual charateristics of age, MMSE, and UFOV performance. ROI analysis results of the parametric analysis were derived from this data.
(XLSX)

**S3 File. NIfTI image files of group analysis.** Statistically significant clusters of BOLD contrasts which are 2 Hz, 4 Hz, 8 Hz, 8 Hz > 2 Hz, and 4 Hz > 2 Hz in Fig 3 and positive slope and positive and negative offset in Fig 6. These files can be mapped on the averaged brain surface (i.e., fsaverage) using a visualization tool (i.e., Freeview) in FreeSurfer.
(ZIP)

## Acknowledgments

We thank Enago for the English language review and the individuals who participated in this study.

## Author Contributions

**Conceptualization:** Yuji Uchiyama, Hiroyuki Sakai, Takafumi Ando, Norihiro Sadato.

**Data curation:** Yuji Uchiyama, Takafumi Ando, Norihiro Sadato.

**Formal analysis:** Yuji Uchiyama, Hiroyuki Sakai.

**Investigation:** Yuji Uchiyama, Hiroyuki Sakai, Takafumi Ando, Norihiro Sadato.

**Methodology:** Yuji Uchiyama, Hiroyuki Sakai, Takafumi Ando, Norihiro Sadato.

**Project administration:** Yuji Uchiyama, Takafumi Ando, Norihiro Sadato.

**Resources:** Yuji Uchiyama, Takafumi Ando, Norihiro Sadato.

**Software:** Yuji Uchiyama, Hiroyuki Sakai.

**Supervision:** Yuji Uchiyama, Takafumi Ando, Norihiro Sadato.

**Validation:** Hiroyuki Sakai, Norihiro Sadato.

**Visualization:** Yuji Uchiyama.

**Writing – original draft:** Yuji Uchiyama, Hiroyuki Sakai, Takafumi Ando, Atsumichi Tachibana, Norihiro Sadato.

**Writing – review & editing:** Yuji Uchiyama, Hiroyuki Sakai, Takafumi Ando, Atsumichi Tachibana, Norihiro Sadato.

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
