## [Decision Letter · Decision Letter 0]

26 Mar 2021

PONE-D-21-00897

BOLD signal response in primary visual cortex to flickering checkerboard increases with stimulus temporal frequency in older adults

PLOS ONE

Dear Dr. Uchiyama,

Thank you for submitting your manuscript to PLOS ONE. After careful consideration of the reviews by two experts in the field, I feel that, while your paper has considerable merit, it does not fully meet PLOS ONE’s publication criteria as it currently stands. Therefore, I would like to invite you to submit a revised version of the manuscript that addresses the points raised during the review process.

The first reviewer (R1) found that the rationale for the study was presented clearly but suggested that clarification with respect to how the methods of the current study could resolve the inconsistent findings you highlight in your review of the literature. R1 also posed some questions, specifically regarding how you assured that participants maintained attention on the stimuli (R2 also raised this point), ROI validity, your modeling and individual differences that are apparent in your results.  Finally R1 suggested some additional considerations for your interpretation of results.  These considerations, along with useful suggestions R1 made regarding additional analyses should be carefully considered in your revision. 

The second reviewer (R2) pointed at some inaccuracy in how you described your result of principal interest (Cliff et al., 2013).  R2 encouraged more careful framing of the problem at hand based on a carefully considered description of the Cliff results.  In this same vein, R2 was concerned about your interpretation of results insofar as your study does not feature age-differential comparisons.  You must be sure to be careful in characterizing your results with respect to age comparisons when none in fact have been made.  Like R1, R2 asked for more detail regarding the descriptions of your analyses.  Finally, R2 also pointed out variance between your results-reporting and the PLOS ONE SAMPL guidelines.

It is clear that the reviewers carefully read and constructively criticized your work.  I strongly encourage you to revise your manuscript along the lines they have provided.

I look forward to receiving your revised manuscript.

Best regards,

Bart Rypma

Academic Editor

PLOS ONE

Journal Requirements:

2. Please note that PLOS ONE does not copy edit accepted manuscripts (https://journals.plos.org/plosone/s/criteria-for-publication#loc-5). To that effect, please ensure that your submission is free of typos and grammatical errors.

*In line with PLOS' guidelines on detailed reporting (https://journals.plos.org/plosone/s/criteria-for-publication#loc-3), please ensure that you have provided sufficient detail on participant recruitment in the Methods section.

"Toyota Central R&D Labs., Inc. (TCRDL) received annual research funding from its

stockholder companies (Toyota Industries Corporation, Toyota Motor Corporation,

Aichi Steel Corporation, JTEKT Corporation, Toyota Auto Body Co., Ltd., Toyota

Tsusho Corporation, Aisin Seiki Co., Ltd., Denso Corporation, Toyota Boshoku

Corporation). This study was performed as part of the self-motivated research activities

in TCRDL. Thus, the funders had no role in the study design, data collection and

analysis, and the decision to publish."

We note that one or more of the authors is affiliated with the funding organization, indicating the funder may have had some role in the design, data collection, analysis or preparation of your manuscript for publication; in other words, the funder played an indirect role through the participation of the co-authors. If the funding organization did not play a role in the study design, data collection and analysis, decision to publish, or preparation of the manuscript and only provided financial support in the form of authors' salaries and/or research materials, please do the following:

a) Review your statements relating to the author contributions, and ensure you have specifically and accurately indicated the role(s) that these authors had in your study. These amendments should be made in the online form.

b) Confirm in your cover letter that you agree with the following statement, and we will change the online submission form on your behalf:

4 Thank you for stating the following in the Competing Interests section:

"I have read the journal’s policy, and the authors of this manuscript have the following

competing interests: YU, HS, TA, and AT were employed by Toyota Central R&D

Labs., Inc. The authors have not applied for any patent related to this study."

Reviewers' comments:

Reviewer's Responses to Questions

**Comments to the Author**

1. Is the manuscript technically sound, and do the data support the conclusions?

Reviewer #1: Yes

Reviewer #2: Yes

2. Has the statistical analysis been performed appropriately and rigorously? 

Reviewer #1: Yes

Reviewer #2: Yes

3. Have the authors made all data underlying the findings in their manuscript fully available?

Reviewer #1: Yes

Reviewer #2: Yes

4. Is the manuscript presented in an intelligible fashion and written in standard English?

Reviewer #1: Yes

Reviewer #2: Yes

5. Review Comments to the Author

Reviewer #1: This study is an analysis of BOLD response in primary visual cortex during basic visual processing in a sample of older adults. Prior literature is mixed on whether or not older adults show increased BOLD response in response to increasing visual frequency rates, and the authors aim to test this by using surface-based analyses which may be more sensitive to anatomical specifics of primary visual cortex. The paper is well written and the analyses straightforward. My comments primarily regard clarification on some of the methods/analysis and suggestions to explore potential individual difference effects.

1. Introduction: The authors do a good job outlining the inconsistent findings from previous studies and possible reasons for the inconsistencies. I think the last paragraph of the Intro would also benefit from more explicit statements about how their study design/methods (e.g., surface based analyses) can help to resolve prior inconsistent findings.

2. Methods: In the discussion the authors mention the task was completely passive viewing, but were any study procedures implemented to ensure that participants remained awake/had their eyes open during the checkerboard task? A few examples of what I mean: eye-tracking, or a camera positions on the participant’s eyes that the experimenter could monitor, or a post-scan debrief asking if the participant remained awake?

3. Methods: FreeSurfer’s segmentation of the cortical surface can sometimes be problematic, especially at the back of the head where dura is frequently included within the surface boundary of posterior occipital regions. Given the importance of this surfaced-based analysis, where the FreeSurfer surfaces checked or edited to ensure no dura matter was included in the primary regions of interest?

4. Methods: For the first-level individual level analysis, why did the authors choose to model each frequency separately (i.e., three regressors) rather than having one parametric regressor that modeled the different levels of frequency (2,4,8 as weights)? A first-level parametric regressor may be more appropriate if the authors anticipate that the change in BOLD response in response to change in frequency was a dose-response type effect (e.g., BOLD showed a greater increase from 4hz to 8hz compared to 2hz to 4hz).

5. Results: Although there is a group effect of increasing frequency being associated with increased BOLD response, only 16 out of 29 participants showed this pattern at the individual level (for at least one hemisphere). Where there any characteristics of the participants that did not show this effect that might explain this (ie., Were they the oldest participants? did they have worse vision? Were they less attentive during scanning?). It would be quite interesting if the authors could quantify the amount of BOLD increase in response to frequency (perhaps through a first-level parametric regressor as suggested above) and link it to individual difference measures (e.g., visual acuity, visual speed of processing).

6. Discussion: Given that the authors find this increase in BOLD in response to increasing frequency in older adults (as is found in younger adults), might this suggest that the difficulties older adults have with visual stimuli are occurring further along the visual processing stream in higher level visual areas? Or perhaps top-down processes are interfering with visual processing? As stated above, given the somewhat large individual differences in how many older adults showed this specific pattern of frequency related BOLD increase, it would be interesting to see if this was predictive of some kind of visual processing behavioral outcome (which might help understand if visual deficits are happening further down the processing stream).

Reviewer #2: The article by Uchiyama and colleagues submitted for consideration to PLOS One examined BOLD signal changes in older (but not younger) adults in visual cortex in response to flickering checkerboard stimuli, the frequencies of which were parametrically varied between 2, 4, and 8 Hz. Results indicated that, contrary to results reported in Cliff et al. (2013), BOLD signal in fact increases with increasing flicker frequency in older adults, in similar fashion to the pattern widely observed in younger adults, and found no part of visual cortex where BOLD signal at 2 Hz was higher than at 8 Hz.

Overall, the article is well-written, though it may benefit from language editing in a few select places (Enago did a reasonably good job, but it can be improved further). I do have several concerns that I believe, if addressed, will significantly strengthen the manuscript.

▪ The central premise of the article seems to be motivated by a single 2013 study by Cliff et al. that showed "decreasing BOLD signal" with increasing flicker frequency. Because it plays so prominently in the rationale behind the study's conception, I would request that the reasoning be more thoroughly fleshed out.

▫ The Cliff et al. (2013) study does not actually show decreasing BOLD signal in older adults with increasing checkerboard flicker frequency, despite how Figure 4 from that article frames the result. The actual quantity depicted is something called the SSQRatio, which is a ratio of sums of squares, and thus does not account for degrees of freedom. This makes interpretation of any differences in the quantity much less straightforward than those of a traditional/parametric test statistic, and in fact, the authors themselves refer to the statistic itself as a goodness-of-fit statistic. Additionally, while the Cliff et al. article does mention that a significant interaction effect between age group and presentation rate was observed for the SSQRatio in bilateral primary visual cortex, it curiously does not provide any statistical metadata regarding that result.

▫ Other concerns with how the authors from the Cliff et al. article processed the imaging data seem to be sufficient to me to question the result (taken together with the first and second points above). Rather than substantially rewriting the discussion or spending unwarranted manuscript real estate to denigrate another article, I would encourage the authors to reframe the introduction to be less dependent upon that article in motivating the study. I believe the authors of the present article got the treatment of Cliff et al. right in the Discussion section, where it is far less featured.

▪ I am also curious as to the authors' choice of sample for the study. The interest was very obviously in potential effects of age on the BOLD signal in response to visual stimuli, but knowing that the Cliff et al. study could serve as a (however imperfect) model, why were younger participants not sampled as well? This is not a fatal flaw in design; far from it. But as no data for younger adults are presented here, I would ask the authors to either remove or significantly reword the last sentence from the abstract, as without a younger group, the question of how age affects V1 response is something this study cannot address. The Discussion paragraph that speaks to this (lines 287-292) begins in more appropriately worded fashion, although the last sentence should probably be softened as well, perhaps even by changing "almost the same" to "similar".

▪ The Materials and Methods section mentions convolution of a boxcar function with a hemodynamic response function. I would request the authors to be more specific here. Which one? Gamma? Double-gamma (similar to the SPM "canonical"? The choice of HRF is very important, as pointed out in Lindquist et al. (2009), especially knowing that older adults were scanned. See West et al. (2019) for concerns regarding changes with age to the BOLD-HRF.

▪ The Materials and Methods section mentions that FreeSurfer/FsFast was used for neuroimaging data analysis, but I was not clear whether this was used for post-hoc statistical testing as well. I would request the authors to be more specific here.

▪ Additional opportunities for language editing did appear from time to time as I read through this manuscript. For example, the first sentence of the ROI analysis subheading of the Materials and Methods section (line 158), sounds more correct as: "To analyze overall V1 activity, ROI analyses were conducted." As I said above, minor changes, but they could be impactful nonetheless.

▪ The authors' reporting of statistical test outcomes in the Results section does not conform to the "SAMPL" guidelines listed in PLOS One's author guidance. Per those SAMPL guidelines, "Avoid relying solely on statistical hypothesis testing, such as P values, which fail to convey important information about effect size... P values are not sufficient for re-analysis. Needed instead are descriptive statistics for the variable being compared, including sample size of the groups involved, the estimate (or “effect size”) associated with the P value, and a measure of precision for the estimate, usually a 95% confidence interval."

▫ In reading through the Results section, the authors have indeed provided only p-values. I would request that they provide estimates of effect size and, as their chosen alpha was 0.05, 95% confidence intervals.

▫ Per the SAMPL guidelines, p-values should not be reported as inequalities. It seems to me as though this is done throughout as a reminder when a significant result is disclosed that it is, indeed, significant, but this is unnecessary and potentially confusing. I would request they remove mentions of "< 0.05".

▫ The SAMPL guidelines are perfectly clear as to the use of "NS": it is to be avoided. As the authors provided the actual p-value in the results (p = 0.054), I would request they simply remove the "NS" from the figure and its mention from the caption.

▪ As to the limitation mentioned immediately prior to the Conclusion (lines 293-300), were participants monitored during scanning? If so, this could be mentioned as a point to help offset the potential concern of this limitation for readers.

6. PLOS authors have the option to publish the peer review history of their article (what does this mean?). If published, this will include your full peer review and any attached files.

Reviewer #1: **Yes: **Jenny Rieck

Reviewer #2: **Yes: **Monroe P Turner

---

## [Author Response · Author response to Decision Letter 0]

19 May 2021

Please find the attached file labeled "Response To Reviewers".

---

## [Decision Letter · Decision Letter 1]

26 Aug 2021

PONE-D-21-00897R1

BOLD signal response in primary visual cortex to flickering checkerboard increases with stimulus temporal frequency in older adults

PLOS ONE

Dear Dr. Uchiyama,

Thank you for submitting your manuscript to PLOS ONE. The paper was re-evaluated by two experts on the field, both of them found that the manuscript improved a lot, however, there are several important concerns that still should be addressed.Therefore, we invite you to submit a revised version of the manuscript that addresses the points raised during the review process.

We look forward to receiving your revised manuscript.

Kind regards,

Andrea Antal, PhD

Academic Editor

PLOS ONE

Reviewers' comments:

Reviewer's Responses to Questions

**Comments to the Author**

1. If the authors have adequately addressed your comments raised in a previous round of review and you feel that this manuscript is now acceptable for publication, you may indicate that here to bypass the “Comments to the Author” section, enter your conflict of interest statement in the “Confidential to Editor” section, and submit your "Accept" recommendation.

Reviewer #1: All comments have been addressed

Reviewer #2: (No Response)

2. Is the manuscript technically sound, and do the data support the conclusions?

Reviewer #1: Yes

Reviewer #2: Yes

3. Has the statistical analysis been performed appropriately and rigorously? 

Reviewer #1: Yes

Reviewer #2: Yes

4. Have the authors made all data underlying the findings in their manuscript fully available?

Reviewer #1: Yes

Reviewer #2: Yes

5. Is the manuscript presented in an intelligible fashion and written in standard English?

Reviewer #1: Yes

Reviewer #2: Yes

6. Review Comments to the Author

Reviewer #1: (No Response)

Reviewer #2: The resubmitted article by Uchiyama and colleagues examined BOLD signal changes in older (but not younger) adults in visual cortex in response to flickering checkerboard stimuli, the frequencies of which were parametrically varied between 2, 4, and 8 Hz. Results indicated that, contrary to results reported in Cliff et al. (2013), BOLD signal in fact increases with increasing flicker frequency in older adults, in similar fashion to the pattern widely observed in younger adults, and found no part of visual cortex where BOLD signal at 2 Hz was higher than at 8 Hz.

The authors, by and large, did an excellent job in responding to the concerns I raised in my initial review: the study is more appropriately motivated without undue reliance on the Cliff et al. result, methods were made more detailed, and results were brought in line with SAMPL guidelines. A few manageable but very important concerns remain, however, and I believe the manuscript will be acceptable for publication if these can be addressed.

• Regarding the choice of the HRF (previous comment 2-4), one critique that remains is the use of a canonical HRF in older populations. Lindquist et al. (2009) elegantly make the point that the choice of HRF used to model measured BOLD signal has important consequences for results and their interpretation. The authors chose the canonical HRF from SPM, a double-gamma function widely employed in the analysis of fMRI data. However, as pointed out in West et al. (2019), this function makes rigid assumptions regarding evolution of the signal’s time-course (e.g., the canonical HRF’s time-to-peak is not free to vary, despite the finding in that study that it is significantly greater in older than in younger adults). This is especially true given that temporal and dispersion derivatives were not included. To be clear, I am not suggesting the authors reprocess their data an additional time using a different HRF selection (though I would consider it to be one way to resolve this criticism). However, in the absence of that step, I do believe that this should be discussed as an additional limitation (in the Discussion section) in reasonable detail for the rationale expounded above.

N.B. My apologies to the authors, in looking at my previous review, the citations I mentioned in comment 2-4 were cut off from the end of the review, and so I have ensured that they now appear in this version.

Lindquist, M. A., Loh, J. M., Atlas, L. Y., & Wager, T. D. (2009). Modeling the hemodynamic response function in fMRI: efficiency, bias and mis-modeling. NeuroImage, 45(1), S187-S198.

West, K. L., Zuppichini, M. D., Turner, M. P., Sivakolundu, D. K., Zhao, Y., Abdelkarim, D., ... & Rypma, B. (2019). BOLD hemodynamic response function changes significantly with healthy aging. NeuroImage, 188, 198-207.

• In the Introduction, the beginning the new third paragraph (beginning “The difficulty of seeing brief…”) ostensibly frames the expectation that neural responses in aged primary visual cortex, by way of reduced information processing capability, actually decrease with increasing temporal frequency. This is later confirmed explicitly in the next paragraph. While I believe this to be a reasonable and logical hypothesis based on what is presented in the Introduction, I was surprised to see no Discussion real estate whatsoever expended to explain the fact that the study finds the complete opposite. If the hypothesis was that BOLD signal will decrease monotonically with increasing checkerboard frequency, what would explain the observation that the opposite is actually true? I believe the authors should discuss this as best they can based on the analysis and methodology they used (and in the context of other work - I understand and applaud not wanting to speculate beyond data they observed).

• The authors have expanded on the rationale for utilizing surface-based analysis methods over volumetric methods. I agree with this rationale, and would suggest citing more literature (than just Brodoehl et al., 2020) to have an empirical basis upon which to base the expectation that surface-based methods will be superior to volumetric methods used in prior work. Specifically, I would like to see the citations below included.

Hutchison, J. L., Hubbard, N. A., Brigante, R. M., Turner, M., Sandoval, T. I., Hillis, G. A. J., ... & Rypma, B. (2014). The efficiency of fMRI region of interest analysis methods for detecting group differences. Journal of neuroscience methods, 226, 57-65.

Tucholka, A., Fritsch, V., Poline, J. B., & Thirion, B. (2012). An empirical comparison of surface-based and volume-based group studies in neuroimaging. NeuroImage, 63(3), 1443-1453.

• I found the description of the Visual processing speed/UFOV task to be lacking. The authors cited a 2017 article which does describe the task very well, which is fine, but I couldn’t quite discern what the second sentence was saying in the context of this manuscript. I don’t expect the authors to describe the task to the same extent as the 2017 paper, but a little more elaboration/clarification would be appreciated here.

• Finding negative BOLD signal relationships with the offset contrast (i.e., the general/frequency-independent effect of stimulus) with no such negative relationships with the slope contrast was interesting. The manuscript indicates that it was in line with previous studies, and cites [43-46], but no such entries appear in the References section. Please add these. Beyond that, though, I felt the (sixth) Discussion paragraph that asserts this as evidence for an experiment and analyses that were well-conducted was lacking. I feel that readers may not follow exactly why this result reinforces the experimental and analytical appropriateness. Please elaborate here (e.g., spatial nature of V1? Anterior/posterior?). I would also like to echo an earlier point – the finding of absolutely no significant clusters that showed a negative relationship to the slope contrast goes completely against the hypothesis laid out in the Introduction, so substantive discussion of one or more potential reasons is warranted.

Minor Point: Table 3 could use a little cleaning up visually (e.g., the word “Positive” is misspelled, the cells are only merged across columns within rows but gray row dividers are still visible).

7. PLOS authors have the option to publish the peer review history of their article (what does this mean?). If published, this will include your full peer review and any attached files.

Reviewer #1: **Yes: **Jenny Rieck

Reviewer #2: **Yes: **Monroe P Turner

---

## [Author Response · Author response to Decision Letter 1]

10 Sep 2021

Please find the attached file named "Response To Reviewers".

---

## [Decision Letter · Decision Letter 2]

18 Oct 2021

BOLD signal response in primary visual cortex to flickering checkerboard increases with stimulus temporal frequency in older adults

PONE-D-21-00897R2

Dear Dr. Uchiyama,

We’re pleased to inform you that your manuscript has been judged scientifically suitable for publication and will be formally accepted for publication once it meets all outstanding technical requirements.

Kind regards,

Andrea Antal, PhD

Academic Editor

PLOS ONE

Additional Editor Comments (optional):

Reviewers' comments:

Reviewer's Responses to Questions

**Comments to the Author**

1. If the authors have adequately addressed your comments raised in a previous round of review and you feel that this manuscript is now acceptable for publication, you may indicate that here to bypass the “Comments to the Author” section, enter your conflict of interest statement in the “Confidential to Editor” section, and submit your "Accept" recommendation.

Reviewer #1: All comments have been addressed

Reviewer #2: All comments have been addressed

2. Is the manuscript technically sound, and do the data support the conclusions?

Reviewer #1: Yes

Reviewer #2: Yes

3. Has the statistical analysis been performed appropriately and rigorously? 

Reviewer #1: Yes

Reviewer #2: Yes

4. Have the authors made all data underlying the findings in their manuscript fully available?

Reviewer #1: Yes

Reviewer #2: Yes

5. Is the manuscript presented in an intelligible fashion and written in standard English?

Reviewer #1: Yes

Reviewer #2: Yes

6. Review Comments to the Author

Reviewer #1: (No Response)

Reviewer #2: (No Response)

7. PLOS authors have the option to publish the peer review history of their article (what does this mean?). If published, this will include your full peer review and any attached files.

Reviewer #1: **Yes: **Jenny Rieck

Reviewer #2: **Yes: **Monroe P Turner